# Federated GNNs for EEG-Based Stroke Assessment

**Andrea Protani**[*♮]    **Lorenzo Giusti**[*]    **Albert Sund Aillet**[*]

**Simona Sacco**[†]    **Paolo Manganotti**[♭]    **Lucio Marinelli**[♯]    **Diogo Reis Santos**[*]

**Pierpaolo Brutti**[♮]    **Pietro Caliandro**[¶]    **Luigi Serio**[*]

**Editors:** Marco Fumero, Clementine Domine, Zorah Lähner, Donato Crisostomi, Luca Moschella, Kimberly Stachenfeld

## Abstract

Machine learning (ML) has the potential to become an essential tool in supporting clinical decision-making processes, offering enhanced diagnostic capabilities and personalized treatment plans. However, outsourcing medical records to train ML models using patient data raises legal, privacy, and security concerns. Federated learning has emerged as a promising paradigm for collaborative ML, meeting healthcare institutions' requirements for robust models without sharing sensitive data and compromising patient privacy. This study proposes a novel method that combines federated learning (FL) and Graph Neural Networks (GNNs) to predict stroke severity using electroencephalography (EEG) signals across multiple medical institutions. Our approach enables multiple hospitals to jointly train a shared GNN model on their local EEG data without exchanging patient information. Specifically, we address a regression problem by predicting the National Institutes of Health Stroke Scale (NIHSS), a key indicator of stroke severity. The proposed model leverages a masked self-attention mechanism to capture salient brain connectivity patterns and employs EdgeSHAP to provide post-hoc explanations of the neurological states after a stroke. We evaluated our method on EEG recordings from four institutions, achieving a mean absolute error (MAE) of 3.23 in predicting NIHSS, close to the average error made by human experts (MAE $\approx$ 3.0). This demonstrates the method's effectiveness in providing accurate and explainable predictions while maintaining data privacy.

## 1   Introduction

Neurological evaluation involving brain signals is the primary tool in assessing and managing stroke patients [1, 2, 3]. These evaluations provide insights into the functional state of biological neural networks affected by stroke, which are essential for guiding clinical decisions and rehabilitation strategies [4, 5, 6]. However, the susceptible nature of such data poses significant challenges, particularly regarding privacy and security [7]. These challenges often hinder data sharing across institutions, limiting the collaborative potential for advancements in clinical neuroscience research and the development of robust predictive models [8, 9].

[*]CERN, [†] University of L'Aquila, [♭]University of Trieste, [♯]University of Genoa,[♮]Sapienza University, [¶]Policlinico Universitario Agostino Gemelli. Correspondence: `andrea.protani@cern.ch`

Proceedings of the II edition of the Workshop on Unifying Representations in Neural Models (UniReps 2024).

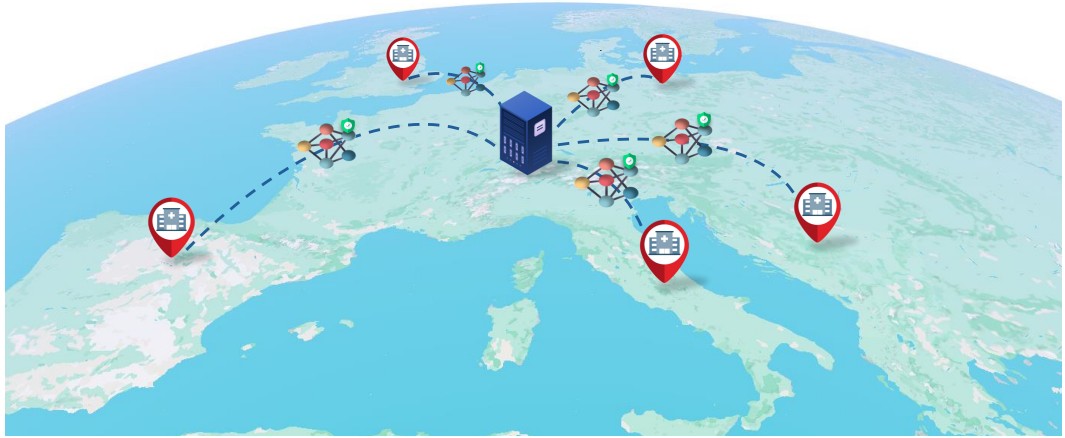

Figure 1: Illustration of a FL setup with hospitals acting as nodes. Each hospital processes data locally while sharing model updates (represented by arrows) with a central server.

**Federated learning (FL)** has emerged as a promising solution to these challenges by enabling multiple institutions to collaboratively train models without sharing sensitive data (Fig. 1) [10, 11]. This approach preserves privacy while leveraging the collective power of diverse datasets. This work proposes a **federated learning framework** based on the Message Queuing Telemetry Transport (MQTT) communication protocol to predict stroke severity using Graph Neural Networks (GNNs) on brain networks extracted from EEG data. GNNs are particularly suited for this task due to their ability to model complex relationships within brain networks, capturing both structural and functional connectivity patterns to provide a better assessment of the stroke impact on brain regions [12, 13, 14, 15]. Moreover, the integration of **explainability mechanisms** enhances the interpretability of the model, which is of primary importance in clinical settings where understanding the rationale behind networks' decisions is critical.

**Dataset and Preprocessing**

This study utilizes a comprehensive dataset comprising EEG recordings from 72 patients collected during hospitalization across four medical centers. The EEG data were analyzed across various frequency bands to construct brain connectivity graphs. The distribution of patients among the hospitals is shown in Fig. 2. A standardized data collection protocol was implemented across all hospitals. EEG signals were recorded at rest, with patients' eyes closed, for at least 5 minutes during their stay in the stroke unit. The recordings were obtained using 31 electrodes positioned according to the international $10 - 10$ system, with a common reference electrode placed on the mastoid and a ground electrode.

The EEG data were preprocessed with a band-pass filter between $0.2$ and $47$ Hz at a $512$ Hz sampling rate, followed by artifact removal through Independent Component Analysis (ICA) [16]. Next, eLORETA [17] was employed to reconstruct whole-brain sources and calculate Lagged Linear Coherence (LLC) graphs for the first five frequency bands: $\delta$ ($2 - 4$ Hz), $\theta$ ($4 - 8$ Hz), $\alpha_1$ ($8 - 10.5$ Hz), $\alpha_2$ ($10.5 - 13$ Hz), and $\beta_1$ ($13 - 20$ Hz) [4].

The NIH Stroke Scale (NIHSS) is an objective assessment tool to quantify the impact of stroke-related events. Initially intended to evaluate the effects of intervention in clinical trials, it has since become a standard tool in clinical and emergency settings for assessing stroke severity. It is an integer value between $1$ and $42$, which quickly determines the level of neurological impairment and guides treatment decisions by systematically evaluating factors such as consciousness, visual fields, motor function, and language ability.

Despite the growing interest in applying graph theory to brain graphs in the context of neuroscience [18, 19, 20, 21], existing approaches often fall short in addressing the unique challenges posed by EEG data. Traditional models lack the structured inductive bias required to capture the connectivity patterns inherent in EEG data [22, 23, 24, 7]. Our approach addresses these limitations and ensures that collaborating institutions maintain data privacy [25, 26, 27]. In particular, our

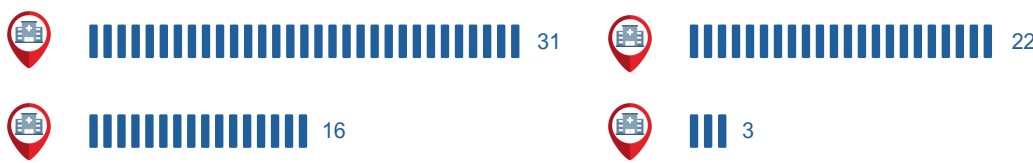

Figure 2: Number of patients for each hospital in our federation.

approach leverages explicitly a **multilayer GNN** to capture insights from graphs extracted from the five different EEG frequency bands defined above. The multilayer network representation allows for integrating information from various neural oscillatory patterns, enabling the model to learn interactions across different frequency bands and enhancing its ability to predict stroke severity.

**Contributions**    In this work, we design a federated learning framework building upon recent work on Multilayer Dynamic Graph Attention Networks for neurological assessments [28] to predict stroke severity from EEG signals in a privacy-preserving setup. This enables multiple healthcare institutions to unify multiple graph representations. We evaluate both FedAvg [10] and SCAFFOLD [29] algorithms, comparing their performance in handling the statistical heterogeneity inherent in multi-institutional EEG data. Our federated learning system integrates MQTT as an efficient communication protocol, demonstrating its security in dispatching model updates and aggregation across distributed clients. We validate our method approach on a dataset of EEG recordings from 72 stroke patients across multiple institutions, demonstrating its effectiveness in providing accurate predictions while ensuring data privacy and robustness. Finally, we provide interpretable explanations of model decisions using EdgeSHAPer [30] to allow for transparent and trustworthy decisions of our predictive framework.

## 2    Background and Related Works

This section reviews the concepts of GNNs and FL and discusses previous studies related to EEG-based stroke assessment.

### 2.1    Graph Neural Networks

GNNs operate on graph-structured data, where a graph $\mathsf{G} = (\mathsf{V}, \mathsf{E})$ consists of nodes $v \in \mathsf{V}$ and edges $(u, v) \in \mathsf{E}$. Each node $v$ is associated with a feature vector $\mathbf{x}_v \in \mathbb{R}^d$. GNNs can be formally defined as functions of the form $\mathrm{GNN}_{\mathbf{w}} : (\mathsf{G}, \{\mathbf{x}_v\}_{v \in V}, \mathbf{w}) \mapsto y_{\mathsf{G}}$, where $\mathbf{w}$ represents a set of trainable parameters. The core idea behind GNNs is to learn node or graph-level representations by iteratively updating node features through message passing:

$$\mathbf{h}_v^{l+1} = \mathrm{com}\left(\mathbf{h}_v^l, \underset{u \in \mathcal{N}(v)}{\mathrm{agg}}\left(\mathsf{M}\left(\mathbf{h}_u^l, \mathbf{h}_v^l, \mathbf{h}_{uv}\right)\right)\right), \tag{1}$$

where $\mathbf{h}_v^l$ represents the hidden feature vector of $v$ at layer $l$, and $\mathbf{h}_v^0 = \mathbf{x}_v$, while $\mathcal{N}(v)$ is the set of neighbors of $v$. For the edge between $u$ and $v$, $\mathbf{h}_{uv}$ represents its hidden feature vector. The function com updates the hidden feature vector with messages received from $u \in \mathcal{N}(v)$, agg is a permutation-invariant aggregation function and $\mathsf{M}$ is a message function.

**Graph Attention Networks**    GATv2 [31] introduces a self-attention mechanism to account for the varying importance of neighboring nodes during message passing. For a node $v$ and its neighbor $u \in \mathcal{N}(v)$, the attention score $s_{\mathbf{w}}(\mathbf{x}_v, \mathbf{x}_u)$ is computed as $s_{\mathbf{w}}(\mathbf{x}_v, \mathbf{x}_u) = \mathbf{a}^\top \sigma\left(\mathbf{W}\left[\mathbf{x}_v \parallel \mathbf{x}_u\right]\right)$, where $\mathbf{W}$ is a learnable weight matrix, $\mathbf{a}$ is a learnable vector of attention coefficients, $\sigma$ is the LeakyReLU activation function and $\parallel$ denotes concatenation. The attention scores are then normalized via $\alpha_{v,u} = \mathrm{softmax}_{u \in \mathcal{N}(v)}\left(s_{\mathbf{w}}(\mathbf{x}_v, \mathbf{x}_u)\right)$.

### 2.2    Multi-layer Brain Networks

To represent brain connectivity across different frequency bands, a multi-layer graph approach has been proposed [28]. The multi-layer graph $\bar{\mathsf{G}}$, illustrated in Fig. 3, is defined as $\bar{\mathsf{G}} = \{\mathsf{G}_{\alpha_1}, \mathsf{G}_{\alpha_2}, \mathsf{G}_{\beta_1}\}$, where each $\mathsf{G}_i = (\mathsf{V}, \mathsf{E}_i)$ represents a graph for a specific frequency band, sharing the same vertices

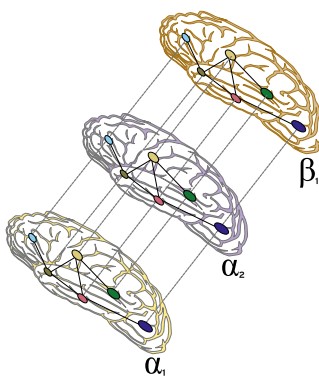

Figure 3: Illustration of the multi-layer network structure. Each layer corresponds to a specific frequency band, with inter-layer connections helping cross-frequency information flow [28].

$V$ but with unique edge sets $E_i$. Inter-layer edges $E_{inter}$ connect nodes across different layers, allowing cross-frequency communication.

**Rewiring Brain Networks**  To address the issue of over-smoothing [32, 33] in fully connected graphs when using GNNs, a rewiring strategy has been developed [28]. This process transforms the graph $G_l = (V, E_l)$ of each brain network layer into a sparse graph $G'_l = (V, E'_l)$ through structural and functional rewiring.

(a) **Structural Rewiring**: a spatial proximity function $\phi : V \times V \to \mathbb{R}^+$ is defined based on Euclidean distance between Brodmann areas. For each node $v \in V$, the $k = 3$ spatially closest nodes are selected.

(b) **Functional Rewiring**: a function $\psi_l : E_l \to \mathbb{R}$ is defined, mapping each edge to its LLC value in layer $l$. Edges above the $99^{th}$ percentile of LLC values are retained. The final set of edges in the rewired graph for each layer is $E'_l = E_\phi \cup E_\psi \cup \{(v,v) : v \in V\}$, where $E_\phi$ represents structural connections, $E_\psi$ represents functional connections, and self-loops are included.

This rewiring strategy reduces edge density (retaining only $\approx 5\%$ of the initial number of edges) while preserving critical functional and structural information, enhancing the GNN's ability to learn from the underlying brain network topology.

The proposed GNN architecture, combined with the multi-layer graph representation and rewiring strategy, allows effective learning from the complex, sparse brain networks derived from EEG data.

## 2.3 Federated Learning

FL is a paradigm that enables collaborative model training across a federation of multiple institutions [10, 34]. This approach has gathered interest in domains like digital healthcare, where regulations and ethical considerations often restrict data sharing [8]. Formally, let $\mathcal{D}_{c_i}$ represent the local dataset on client $c_i$ and $\mathcal{W}$ denote the global model parameters. Each client $c_i$ updates the global model by minimizing a local loss function $\mathcal{L}_{c_i}(\mathcal{W}; \mathcal{D}_{c_i})$. The global objective can be expressed as:

$$\min_{\mathcal{W}} \sum_{i=1}^{N} \frac{n_i}{n} \mathcal{L}_{c_i}(\mathcal{W}; \mathcal{D}_{c_i}) \tag{2}$$

where $n_i$ is the number of samples on device $c_i$ and $n = \sum_{i=1}^{N} n_i$ is the total number of samples across all devices.

**Federated Averaging (FedAvg)** is one of the most popular algorithm for FL aggregation [10]. FedAvg aims to address the challenges of statistical heterogeneity among participants in federated systems. The algorithm operates as follows: the server initializes the global model parameters $\mathcal{W}^0$. Then, for each round $r = 1, \ldots, R$, the server *samples a subset of clients* $S_r$ according to a probability distribution $\mathcal{P}$ and sends the current global model $\mathcal{W}^{r-1}$ to each of the clients in the sampled subset. Each client $c_i \in S_r$ *updates the model locally* using their dataset $\mathcal{D}_{c_i}$ by computing $\mathcal{W}^r_{c_i} = \text{OPT}(\mathcal{W}^{r-1}, \mathcal{D}c_i)$, where OPT is the local optimization algorithm (e.g., SGD,

Adam, RMSProp). The clients then *send their updated models* back to the server. The server *aggregates these local models* to update the global model using a weighted average:

$$\mathcal{W}^r = \sum_{c_i \in S_r} \frac{n_{c_i}}{n_{S_r}} \mathcal{W}^r_{c_i} \tag{3}$$

where $n_{c_i} = |\mathcal{D}_{c_i}|$ and $n_{S_r} = \sum_{c_i \in S_r} n_{c_i}$. This process repeats for $R$ rounds, resulting in the final global model $\mathcal{W}^* = \mathcal{W}^R$. FedAvg unifies the representations learned by individual clients on their local datasets into a single global model by aggregating their locally updated models. This process effectively captures patterns and features from each client's data, integrating them into a single model that generalizes across the entire dataset distribution. This way of averaging local models allows FedAvg to unify the collective knowledge of all clients into a single representation while preserving data privacy, resulting in an improved global model without direct access to the client's raw data.

**Stochastic Controlled Averaging for Federated Learning (SCAFFOLD)** is an algorithm proposed in [29] to address the client drift problem in heterogeneous settings. SCAFFOLD introduces control variates to correct for the drift in local updates. thereby aligning the clients' learning processes more closely with the global objective. Initially, the server initializes the global model parameters $\mathcal{W}^0$, the server control variate $c^0$ and sets all client control variates $c^0_{c_i} = 0$ for each client $c_i$. For each round $r = 1, \dots R$, the server samples a subset of clients $S_r$ according to a probability distribution $\mathcal{P}$ and sends the current global model $\mathcal{W}^{r-1}$ along with the server control variate $c^{r-1}$ to the selected clients.

Each client $c_i \in S_r$ initializes their local model with the global model: $\mathcal{W}^{r,0}_{c_i} = \mathcal{W}^{r-1}$. The client then performs $K$ local update steps. At each local step $k = 1, \dots K$, the client updates the model as:

$$\mathcal{W}^{r,k}_{c_i} = \mathcal{W}^{r,k-1}_{c_i} - \eta_l \left( \nabla F_{c_i}\left(\mathcal{W}^{r,k-1}_{c_i}\right) - c^{r-1}_{c_i} + c^{r-1} \right) \tag{4}$$

where $\eta_l$ is the *local learning rate*, $\nabla F_{c_i}\left(\mathcal{W}^{r,k-1}_{c_i}\right)$ is the gradient of the local loss function for client $c_i$, $c^{r-1}_{c_i}$ is the client control variate, and $c^{r1}$ is the server control variate. After completing the local updates, the client computes the model update: $\Delta_i = \mathcal{W}^r_{c_i} - \mathcal{W}^{r-1}$, and updates its control variate as:

$$c^r_{c_i} = c^{r-1}_{c_i} - c^{r-1} + \frac{1}{(\eta_l K)} \Delta_i \tag{5}$$

The client then returns $\Delta_i$, and the updated control variate $c^r_{c_i}$ to the server. Upon receiving the updates from all participating clients, the server aggregates the update to update the global model and the server control variate:

$$\mathcal{W}^r = \mathcal{W}^{r-1} + \frac{\eta_g}{|S_r|} \sum_{c_i \in S_r} \Delta_i \tag{6}$$

$$c^r = c^{r-1} = \frac{1}{|S_r|} \sum_{c_i \in S_r} \left(c^r_{c_i} - c^{r-1}_{c_i}\right) \tag{7}$$

This process repeats for $R$ rounds, resulting in the final global model $\mathcal{W}^* = \mathcal{W}^R$.

SCAFFOLD enhanced model aggregation over FedAvg by utilizing control variates to correct for client drift caused by data heterogeneity. In particular, adjusting local updates with these control variates allows SCAFFOLD to ensure that the federated learning process aligns clients' representation into a unified global model that remains synchronized with the global objective, mitigating undesired effects of non-I.I.D. data distributions due to statistical heterogeneity. Consequently, SCAFFOLD is a theoretically sound algorithm that aims to achieve a more accurate and generalized global model that unifies the collective knowledge of all clients while preserving data privacy. It is also worth emphasizing that if an early stopping mechanism is implemented for FedAvg and SCAFFOLD, the training may end at an earlier round $r' < R$ when certain convergence criteria are met (e.g., the validation loss stops decreasing). In this case, the final global model becomes $\mathcal{W}^* = \mathcal{W}^{r'}$.

## 2.4 Edge Shapley Values for Model Explainability

While the model in [28] effectively predicts stroke severity from EEG data, its interpretability is limited to the noise introduced by random initialization of the attention coefficients $a$. In particular,

the functional communication between brain regions allows clinicians to understand *why* the model makes specific predictions, allowing clinicians to trust and utilize these insights in personalized treatment plans. Traditional attention mechanisms provide some interpretability by highlighting important nodes and edges, but they often lack a theoretically grounded measure of each edge's contribution to the prediction.

To address this gap, we incorporate **EdgeSHAPer** [30, 35], as an edge-centric explanation method based on the Shapley value concept from cooperative game theory [36]. EdgeSHAPer assigns a Shapley value to each edge in the brain network, quantifying its contribution to the model's output. This method uses a principled way to assess the importance of individual neural connections, thereby offering more profound insights into the brain's functional reorganization after a stroke.

EdgeSHAPer estimates the Shapley value $\phi(e_i)$ for each edge $e_i$ in the graph $\mathsf{G}$, representing the average marginal contribution of $e_i$ to the model's prediction over all possible subsets of edges. The Shapley value for an edge is defined as:

$$\phi(e_i) = \frac{1}{|\mathsf{E}|} \sum_{S \subseteq \mathsf{E} \setminus \{e_i\}} \frac{f(S \cup e_i) - f(S)}{\binom{|\mathsf{E}|-1}{|S|}} \tag{8}$$

where $\mathsf{E}$ is the set of all edges in $\mathsf{G}$, $S$ is a subset of edges not containing $e_i$, and $f(S)$ is the model's prediction when only the edges in $S$ are present. The factorial terms account for all possible orderings of edges, ensuring a fair attribution of importance. Computing the exact Shapley values is computationally intractable for graphs with many edges due to the combinatorial number of subsets. Therefore, we employ a Monte Carlo approximation [37]:

$$\phi(e_i) \approx \frac{1}{M} \sum_{k=1}^{M} \left[ f(S_k \cup e_i) - f(S_k) \right], \tag{9}$$

where $M$ is the number of sampled subsets $S_k \subseteq E \setminus \{e_i\}$. We generate these subsets by random sampling, ensuring that each edge's contribution is estimated over diverse contexts.

## 2.5 Related Works

The impact of acute stroke on the topology of cortical networks has been extensively investigated through EEG analysis, revealing significant, frequency-dependent alterations in network properties. Specifically, stroke leads to decreased small-worldness in the $\delta$ and $\theta$ bands and increased small-worldness in the $\alpha_2$ band across both hemispheres, regardless of lesion location [1]. Distinct modifications in functional cortical connectivity due to acute cerebellar and middle cerebral artery strokes have been highlighted, showing different impacts on network architecture and small-world characteristics across various EEG frequency bands, independent of ischemic lesion size [4, 19].

Additionally, research has shown that acute cerebellar and middle cerebral artery strokes distinctly affect functional cortical connectivity, with significant differences in EEG-based network remodeling across $\delta$, $\beta_2$, and $\gamma$ frequency bands, highlighting the unique impact of stroke location on brain network dynamics [19]. The prognostic role of hemispherical differences in brain network connectivity in acute stroke patients has been explored using EEG-based graph theory and coherence analysis. Findings indicate that stroke-induced alterations in network architecture can predict functional recovery outcomes, providing a basis for tailored rehabilitation strategies [24, 23].

The relevance of brain network analysis for stroke rehabilitation has been studied, highlighting the potential of network-based approaches to inform and guide therapeutic interventions in stroke recovery [38]. Dynamic functional reorganization of brain networks post-stroke has been emphasized, providing critical insights into the brain's adaptive mechanisms following a stroke and supporting network analysis to understand structural and functional reorganization [3]. Finally, changes in the contralesional hemisphere following stroke and the implications of the stroke connectome for cognitive and behavioral outcomes have been explored, enhancing our understanding of the complex network dynamics involved in stroke pathology and recovery [2, 39, 20]

## 3 Experimental setup

We designed two distinct federated learning setups to investigate how data distribution affects model performance. The first configuration mimics real-world conditions by treating individual hospitals

as clients in the federation, reflecting the natural data distribution within healthcare systems. We curated three evenly distributed datasets in the second setup, creating a more controlled, idealized scenario. By comparing these two approaches, we assess whether having a representative distribution of the entire dataset at each client significantly impacts the learning outcomes. All experiments were conducted using the same model architecture and training hyper-parameters. We used a batch size of 2 and applied the mean squared error (MSE) loss function. We implemented gradient clipping with a threshold of 10 to avoid exploding gradient issues. The model's parameters were optimized using the Adam optimizer with a learning rate of 0.003 and a weight decay of 0.01 to prevent overfitting. For both setups, we fixed a test set by randomly sampling 11 elements across the clients to be representative of the overall data distribution, ensuring consistency in the evaluation process and reducing the risk of bias. We used $M = 100$ for the Monte Carlo approximation in 9 to estimate Shapley values efficiently.

**Network architecture and Communication Protocol**     The model architecture was built on top of the GATv2 message-passing scheme [31]. The core GNN consisted of 3 GATv2 layers, each with 6 attention heads. To reduce overfitting, we incorporated dropout with a 0.1 probability of randomly deactivating neurons during training and used the ReLU activation function for non-linearity. A mean pooling layer was employed as the readout operation following the final GNN layer.

At the core of federated learning systems lies the need for efficient communication protocols to maintain model performance under resource constraints [10, 40, 41]. Ensuring efficient and secure data exchange among distributed actors of federated systems requires carefully tailoring the protocol for communicating models' parameters among the network nodes. We chose the MQTT protocol instead of HTTP REST for our setup due to superior bandwidth efficiency, lower latency, and low overhead, making it suitable for low-power IoT devices [34]. On top of the aforementioned benefits, MQTT offers one-to-many communication, which is the central point when distributing global models during FL rounds. It also supports large message payloads (with fragmentation if necessary), serialization, and compression. The communication network orchestrates FL processes through MQTT's publish-subscribe mechanism. Clients (healthcare institutions) exchange model parameters and configurations by publishing their local models and downloading global models via specific MQTT topics [42, 43].

This architecture handles both synchronous (all clients wait for each other) and asynchronous (clients act independently of others) FL operations. For the FL orchestration, there are four different setups:

(a) Both the Parameter Server (PS) and the clients operate synchronously, waiting for all clients to finish local model optimization before aggregating the global model.

(b) The PS is synchronous, but clients are asynchronous, continuing local training until they receive a global model update.

(c) Both PS and clients are asynchronous, where the PS aggregates models at regular intervals, regardless of client completion.

(d) Fully decentralized and asynchronous, clients exchange models directly via MQTT without a central PS.

We chose option (a) for our experimental setup since we prioritize stability and consistency in the model aggregation process due to the small data samples. All MQTT exchanges are encrypted via TLS, and the MQTT broker manages local models until training is completed. The flexible architecture supports dynamic scaling based on task complexity and computational resources [44].

**Comparative Analysis:** Our experimental framework spans several learning paradigms, allowing us to comprehensively evaluate the performance of FL in the context of GNNs. The scenarios explored include:

(a) **Realistic FL Setup:** Simulating real-world data distribution among hospital clients.

(b) **Idealized FL Setup:** Using manually curated datasets with equal distribution.

(c) **Centralized Learning:** Training a single model on the entire dataset pooled together.

(d) **Isolated Learning:** Training separate models for each client without collaboration.

This approach allows us to:

Table 1: Mean and standard deviation across five different initializations for each setup. In isolated learning, the statistics are also computed across the different clients.

| Experiment | Setup | Algorithm | MAE |
|---|---|---|---|
| Isolated Learning | Realistic | - | $3.68 \pm 0.26$ |
| | Idealized | - | $3.92 \pm 0.87$ |
| Federated Learning | Realistic | FedAvg | $3.23 \pm 0.06$ |
| | | SCAFFOLD | $3.22 \pm 0.05$ |
| | Idealized | FedAvg | $3.34 \pm 0.08$ |
| | | SCAFFOLD | $3.44 \pm 0.07$ |
| Centralized Learning | - | - | $3.22 \pm 0.12$ |

(a) Examine how data distribution affects FL performance by comparing realistic and idealized setups.

(b) Quantify the benefits of collaborative learning in FL versus training models independently for each client.

(c) Analyze the trade-offs between FL and centralized learning, providing insights into the viability of FL in situations where data sharing is limited.

**Computational Resources and Code Assets** In all experiments we used a machine with an NVIDIA® RTX 3090 GPU with an Intel® Xeon® @ 2.30GHz on Ubuntu 22.04 LTS 64-bit. The model was implemented in PyTorch [45] by building on top of PyTorch Geometric library (MIT license) [46]. PyTorch, NumPy [47], SciPy [48] are BSD licensed, Matplotlib [49] is PSF licensed.

## 4 Experimental result

The performances across different learning setups are summarized in Tab. 1. Results reflect the mean absolute error (MAE) and standard deviation across five initializations. The statistics are also computed across clients in isolated learning setups, while federated and centralized setups leverage collaborative training.

The **Isolated Learning** approach, where models are trained independently for each client, shows weaker performance compared to collaborative methods. In the realistic isolated learning setup, a MAE of $3.68 \pm 0.26$ was obtained, while the idealized isolated setup, though more controlled, resulted in a higher MAE of $3.92 \pm 0.87$. The high variance in the idealized setting suggests that the data distribution across clients introduces inconsistencies in model performance, even when the data covers the full value range. This reflects the challenge of achieving stable results when training independently without data sharing.

**FL** consistently outperformed isolated learning. In the realistic FL setup, **FedAvg** achieved a MAE of $3.23 \pm 0.06$, and **SCAFFOLD** slightly improved with an MAE of $3.22 \pm 0.05$. This indicates the robustness of federated learning in harnessing distributed data while maintaining privacy. The marginal improvement of SCAFFOLD over FedAvg in the realistic scenario suggests that SCAFFOLD's correction of client drift is particularly beneficial when client data distributions closely resemble real-world conditions.

Performance was slightly lower in the **Idealized Federated Learning** setup, which uses a more balanced and mixed data distribution across clients. FedAvg reported MAE of $3.34 \pm 0.08$, while SCAFFOLD had an MAE of $3.44 \pm 0.07$. Although more evenly distributed, the idealized setup may obscure the natural variations in client data in realistic conditions. This could explain why models trained in the idealized setup generalize less effectively to the test set, which likely follows a more realistic data distribution. Thus, the realistic setup mirrors client-specific data patterns, enhancing predictive accuracy.

The **Centralized Learning** setup, where all client data is pooled into a single model, achieved a MAE of $3.22 \pm 0.12$, which is identical to the performance of the **Realistic Federated Learning** setup using the SCAFFOLD algorithm. This highlights that federated learning, when properly configured, can achieve the same level of accuracy as centralized learning without requiring data sharing. Given the privacy constraints in medical settings, federated learning presents a substantial

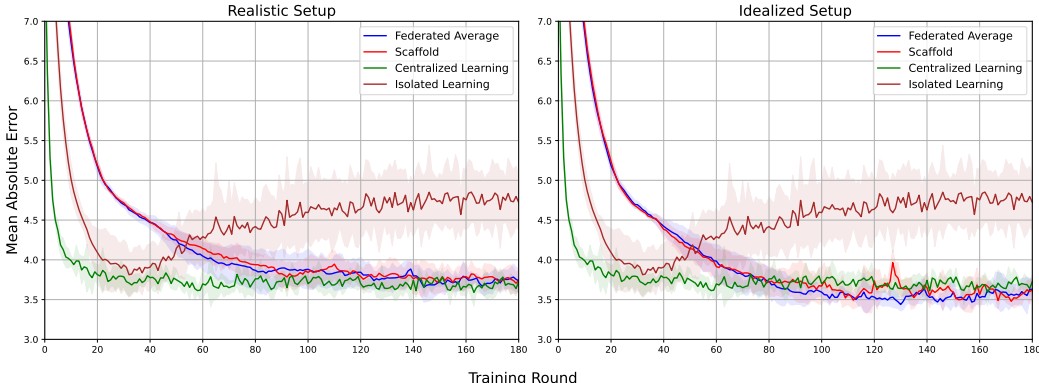

Figure 4: Convergence of MAE over training rounds for different setups. Centralized learning shows the fastest and most stable convergence. The federated learning setups converge more slowly but arrive at the same level as the centralized. The isolated learning approaches exhibit higher error and slower convergence, confirming the benefits of collaborative training in federated setups.

alternative, offering equivalent performance while preserving data locality and complying with privacy regulations. Fig. 4 provides further insights into model convergence across different setups. The plot shows that centralized learning achieves the fastest convergence, as expected, given its access to the full dataset. Among federated approaches, the realistic setup converges as smoothly as the idealized setup.

**Explainable Insights**    Integrating EdgeSHAPer values into our framework enables us to assign a Shapely value to each edge, which offers several key advantages:

(a) **Quantitative Edge Importance**: Shapley values provide a theoretically robust measure of each edge's significance, enabling us to rank physiological connections based on their contributions to the model's predictions. This quantitative assessment ensures a fair and consistent evaluation of edge importance across brain networks.

(b) **Interpretable Visualizations**: We visualize the brain network by coloring edges based on their corresponding Shapley values and sizing nodes according to their weighted degree centrality derived from those values (see Fig. 5). This dual representation effectively highlights critical brain regions and their interconnections, making the network dynamics more comprehensible.

(c) **Actionable Insights**: Identifying the most significant edges allows clinicians to understand which neural pathways were most affected by the stroke. This understanding informs the development of targeted therapeutic interventions, enabling personalized treatment plans that address the specific neural disruptions identified by our model.

We used a similarity metric based on the normalized Euclidean distance to assess the similarity between different model variants. This metric was applied to compare sets of weight vectors derived from the various experiments. Our analysis revealed an average similarity of $0.76$, indicating high coherence among the weight vectors. Notably, the *idealized subgroup* exhibited an average similarity of $0.75$, while the *realistic subgroup* achieved a higher similarity of $0.78$. The similarity matrix demonstrated consistently high values across all vector pairs, ranging from $0.73$ to $0.78$. This approach provides insights into the convergence patterns and consistency of diverse federated learning strategies and setups. Furthermore, these results align with our observations regarding the best MAE, suggesting a strong correlation between higher performance metrics and more cohesive latent representations. Fig. 5 illustrates these findings, depicting the Edge Shapley Values for our federated learning setups and highlighting the relative contributions of different nodes and edges to the model's predictions.

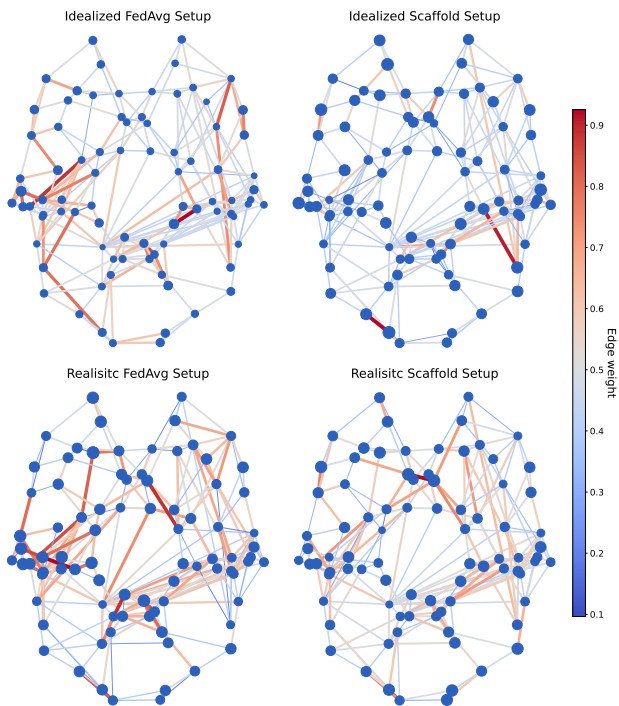

Figure 5: Illustration of Edge Shapley Values for various federated learning setups for the same patient. The color intensity is proportional to the contributions to model predictions. Node sizes are proportional to their weighted degree centrality, adjusted for the number of connections, highlighting each node's significance within the network.

## 5   Conclusions

**Broader Impact**     The proposed federated learning framework for stroke assessment can assist clinical neuroscientists' evaluations by enabling collaborative research and model development across multiple institutions while maintaining strict data privacy. By allowing hospitals to unify insights without sharing raw patient data, this approach addresses privacy concerns and broadens the scope of multi-healthcare collaborations. The model's ability to predict stroke severity with an error rate close to human performance and interpretability modules can help clinicians better understand brain network changes post-stroke, leading to more personalized treatment plans.

**Limitations**     While the proposed federated learning framework offers significant privacy advantages and demonstrates solid predictive performance, some limitations must be acknowledged. First, the relatively small sample size of 72 patients restricts the model's generalizability to more extensive and diverse clinical populations. Additionally, scalability to broader multi-institutional settings may face challenges due to variations in data quality, preprocessing protocols, and hardware capabilities across institutions.

**Conclusion**     This study introduces a novel federated learning framework using GNNs for neurological assessments, demonstrating its ability to predict stroke severity from EEG data across multiple institutions. We show the effectiveness of collaborative model training while preserving data privacy. Incorporating explainability through EdgeSHAPer further enhances the model's potential clinical relevance by providing insights into the neural connections driving predictions. Future work will focus on expanding the dataset and addressing the technical challenges of scaling the system to larger, more diverse populations.

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
