# OpenReview forum: "Federated GNNs for EEG-Based Stroke Assessment"
_NeurIPS.cc/2024/Workshop/UniReps — UniReps_

### Official Review · Reviewer_yQmP · 2024-10-03

**Rating:** 4
**Confidence:** 3

**Review:**

This work introduces a new framework combining Federated Learning (FL) and Graph Neural Networks (GNNs) to predict stroke severity using EEG data. The two topics are based on their respective motivations of privacy sensitiveness and structured learning. The proposed model allows distributed medical institutions to collaboratively train a GNN model on local EEG data without sharing sensitive patient information. The framework integrates explainability through EdgeSHAP and achieves competitive predictive accuracy in stroke severity assessment compared to human experts, all while maintaining data privacy across healthcare institutions.

Strengths:
- the use of federated learning is well motivated and the paper addresses privacy concerns in collaborating hospital using it. although this insight on fed learning is well known
- combining EEG data with GNNs for stroke assessment leverages brain network patterns for accurate predictions.
- the inclusion of EdgeSHAP enhances model explainability, providing insights into which neural connections most influence the predictions, which is crucial for clinical use.
- the framework is evaluated on real-world EEG datasets from multiple institutions.

Limitations:
- there is no rewiring strategy in GATv2 as indicated in line 86, or in the reference [28] to my best reading. the rewiring terminology could refer to sparsification instead. The use of the term 'rewiring' could be a misinterpretation of sparsification.
- the sentence in lines 95-96 'This process transforms each layer Gl = (V, El ) into a sparse graph G′l = (V, E′l ).. ' may be miswritten since 'a layer' is said to be 'transformed to a graph'. Or if it is indeed rewiring then perhaps the writing may appear confusing.
- equation 1 is not properly defined - there is no indication of what 'com' denotes. also additional terms are being introduced without substantial purpose such as h_{new} where 'new' may not be required and is also not defined if it is necessarily introduced.
- minor comment on fig 1 - the figure may not have been attributed appropriately.
- the focus of this work may not be contributing enough on 'unifying representations' despite the paper mentioning this at lines 61-62. the purpose of fed learning may not be unifying representations but instead allow collaborative learning or privacy preserving learning across data or server nodes.

---

### Official Review · Reviewer_m2Ki · 2024-10-04
**The paper offers a well-written and structured study on predicting stroke severity using a novel Federated Learning approach with GNNs. The authors highlight the importance of data privacy in medical machine learning, enabling multiple institutions to train a shared GNN model on local EEG data without exchanging sensitive patient records.  A key strength is the clear explanation of the experimental setup, involving EEG data from four hospitals. The use of a masked self-attention mechanism and EdgeSHAP for model interpretation enhances both performance and explainability.  The results show that the Federated Learning method significantly improves prediction accuracy over isolated models, with a Mean Absolute Error of 3.23, close to the 3 achieved by human experts. However, the paper could benefit from including a relative measure, like percentage improvement, to better quantify the method’s performance gains over baselines.**

**Rating:** 7
**Confidence:** 3

**Review:**

The paper presents a well-structured study on predicting stroke severity using a novel Federated Learning approach with GNNs. The authors emphasize the importance of data privacy, allowing multiple institutions to collaboratively train a shared GNN model on local EEG data without exchanging patient records, thus preserving privacy while benefiting from pooled knowledge.

The experimental setup is clearly explained, detailing how Federated Learning was implemented across four hospitals. The model uses a masked self-attention mechanism to capture important brain connectivity patterns and EdgeSHAP for post-hoc explanations, enhancing both performance and interpretability.

The paper also includes a comparison between different setups, showing how the Federated Learning method outperforms models trained in isolation at each institution. The reported Mean Absolute Error of 3.23 is close to 3.0 achieved by human experts, highlighting the model’s clinical relevance.

While Mean Absolute Error is effective for evaluating performance, adding a relative measure like percentage improvement would provide a clearer view of the performance gains over baseline models.

---

### Official Review · Reviewer_x3N2 · 2024-10-05
**Revision for Federated GNNs for EEG-Based Stroke Assessment**

**Rating:** 8
**Confidence:** 4

**Review:**

Thank you for all your nice work in Federated Learning!

The manuscript uses an existing Graph Neural Network, designed for stroke assessment on EEG data, together with Federated Learning paradigms to evaluate the predictions on different training cases. Maintaining the privacy of medical data is a very relevant task for patient security reasons and therefore the study is very important for the community.

However, there are some issues, especially regarding the definition of the problem and the learning objective, that should be addressed. All issues are addressed below:

Introduction
The terms "EEG" and "MQTT protocol" are mentioned in the abstract (line 8) and the introduction (line 29) before being properly introduced.

The dataset description and processing details are misplaced in the introduction since the used datasets and preprocessing techniques are methodological choices.

Related Works
There are plenty of references to the study “Towards explainable graph neural networks for neurological evaluation on EEG signals.” (line 60), and the prediction model on EEG signals is based on it. However, the paper cannot be located and verified. Please ensure that all references are properly listed and accessible.

Experimental setup
The paper does not clearly specify how "stroke severity" is being quantified. It is unclear whether the metric used is based on the National Institutes of Health Stroke Scale (NIHSS) scores or another standard metric. There are several different ways of classifying a ‘minor stroke’ (e.g., https://doi.org/10.1161/STROKEAHA.109.572883), for example, and understanding the method used to define what you want to predict is essential for the reader’s understanding of the task. This should be clarified, and if some well-known metric was used, proper citation should be provided.

The paper does not clearly state what is the task of the prediction model, or what is the quantity being inferred. If it is a classification task, what would the classes be? Are them stroke severity levels based on a metric? If it is a regression task on quantities related to measuring stroke severity, would these quantities be?

Experimental results
Could you please explain better the choice of using Mean-Squared Error for assessing model performance, instead of Accuracy or Area Under The Curve (AUC), such as other papers in the area (e.g., https://iopscience.iop.org/article/10.1088/1742-6596/1528/1/012006, https://pubmed.ncbi.nlm.nih.gov/33116187/)?

---

### Official Review · Reviewer_Un27 · 2024-10-07
**Review in federated learning framework combined with GNNs**

**Rating:** 6
**Confidence:** 3

**Review:**

In this paper, the author proposes a federated learning framework combined with graph neural networks (GNNs) to predict stroke severity using EEG signals from distributed medical institutions. And the aim is to preserve patient data privacy while enabling hospitals to collaboratively train a shared GNN model. The approach leverages masked self-attention mechanisms to capture critical brain connectivity patterns and EdgeSHAP for post-hoc explainability of neurological states post-stroke.

Pros:

1. The combination of GNNs and Federated Learning for the EEG-based stroke assessment is novel. And I think that this proposed framework addresses privacy concerns.  2. I believe this paper conducted experiments across four institutions, which can be treated to strengthen the generalizability of the results. 3. Also, it includes strong emphasis on explainability with the inclusion of EdgeSHAP. This gives transparency.


Cons

1. The study highly bases on EEG data from 71 patients and I think we need to consider a larger and more diverse dataset even though we can derive the promising results.  2. The paper only evaluates through Mean Absolute Error (MAE) to stroke severity prediction. I believe including more metrics can provide more generalized evaluation of the model's performance. 3. The paper lacks the theoretical justification to GNN model, why we choose GATv2 rather than other GNN architectures? 4. The visualization in the paper of brain networks lack explanation to their roles in the paper. This would enhance paper's readability if the paper includes more how the visualization is related to stroke recovery and how they can be used in clinical practice.


In summary, I would support this paper but there are lots of concerns need to be improved or further explanation.

---

### Decision · Program_Chairs · 2024-10-10

**Decision:**

Accept

**Comment:**

In light of the positive reviewers' feedback and relevancy of the submission, we are pleased to accept this paper for presentation at UniReps 2024. We kindly ask the authors to incorporate and address the reviewers' suggestions and feedback in the final camera-ready version of the manuscript.